# Exploring and Monitoring Privacy, Confidentiality, and Provider Bias in Sexual and Reproductive Health Service Provision to Young People: A Narrative Review

**DOI:** 10.3390/ijerph19116576

**Published:** 2022-05-27

**Authors:** Andrew G. Corley, Andrea Sprockett, Dominic Montagu, Nirali M. Chakraborty

**Affiliations:** Metrics for Management, 12 West Madison St., Baltimore, MD 21201, USA; andrea@m4mgmt.org (A.S.); dominic@m4mgmt.org (D.M.); nirali@m4mgmt.org (N.M.C.)

**Keywords:** young people, reproductive health services, developing countries, privacy, confidentiality, provider bias

## Abstract

Purpose: Poor privacy and confidentiality practices and provider bias are believed to compromise adolescent and young adult sexual and reproductive health service quality. The results of focus group discussions with global youth leaders and sexual and reproductive health implementing organizations indicated that poor privacy and confidentiality practices and provider bias serve as key barriers to care access for the youth. Methods: A narrative review was conducted to describe how poor privacy and confidentiality practices and provider bias impose barriers on young people seeking sexual and reproductive health services and to examine how point of service evaluations have assessed these factors. Results: 4544 peer-reviewed publications were screened, of which 95 met the inclusion criteria. To these articles, another 16 grey literature documents were included, resulting in a total of 111 documents included in the review. Conclusion: Poor privacy and confidentiality practices and provider bias represent significant barriers for young people seeking sexual and reproductive health services across diverse geographic and sociocultural contexts. The authors found that present evaluation methods do not appropriately account for the importance of these factors and that new performance improvement indicators are needed.

## 1. Introduction

### 1.1. Background

Despite the calls to increase modern contraception availability, there remains enormous unmet need. An estimated 218 million women (aged 15–49) living in low- and middle-income countries (LMICs) in 2019 had an unmet need for modern contraception [1]. In LMICs, unmet contraceptive and sexual and reproductive health (SRH) service needs lead to an estimated 111 million annual unintended pregnancies and 133 million women of reproductive age not receiving care for treatable sexually transmitted infections (STIs). Furthermore, adolescent women aged 15 to 19 have an even higher unmet need compared to all women of reproductive age (43% vs. 24%) [1]. 

Unmet SRH needs in adolescents and young adults (AYAs) can carry particularly serious health consequences. AYAs suffer disproportionately high rates of STIs and HIV [2,3]. In 2009, 41% of all new HIV infections in adults over 15 were among young people aged 15 to 24 years [3]. During adolescence, pregnancy can lead to comparatively greater risks for a number of poor health outcomes for both mother and child. Pregnancy-related complications are the second leading cause globally of death among adolescent girls aged 15 to 19 [4]. Additionally, adolescent girls have higher risks of experiencing unintended pregnancy and pregnancy-related complications, such as obstetric fistula, systemic infection, or postpartum hemorrhage, than older reproductive-age women [5]. Infants born to adolescents are also more likely to experience compromised early-life health outcomes. Babies born to mothers under 20 years of age experience higher rates of low birth weight, preterm birth, and asphyxia than those born to older mothers [5]. Finally, unmet contraceptive care needs in AYAs lead to disproportionately high rates of unsafe abortions. Forty-one percent of all unsafe abortions conducted in LMICs are in young women aged 15–24 [6].

Despite frequently being collectively referred to as “young people”, adolescents and young adults oftentimes find themselves in very different life circumstances [7]. However, given their comparable social standing in many cultural contexts and because cognitive and emotional development have been shown to continue well into individuals’ 20s, young adults share much in common with their adolescent counterparts [7,8,9]. There are similarly important reproductive rights, gender empowerment, and economic development consequences to not meeting both adolescent and young adult SRH needs [10]. In terms of their reproductive rights, AYAs have the same right to bodily autonomy and birth spacing as older adults [10,11,12]. Limitations on SRH service availability infringes on these rights. In many countries, AYA women are less likely to be able to make their own decisions regarding their SRH than older women [13]. Likewise, limitations on SRH service provision to AYAs are often driven by restrictive gender norms that maintain power hierarchies and exert negative health effects on both men and women [14]. Whether viewed as the product or driver of empowerment, ensuring access to high-quality SRH services is inextricably linked to AYA gender empowerment [15]. There are also important economic consequences to unmet contraceptive needs and early pregnancy. At the individual level, early marriage and childbirth oftentimes force girls and young women to leave school early, thereby reinforcing cycles of family poverty and violence [16,17]. At the national level, AYA pregnancy is a key barrier to economic development for many LMICs. Early pregnancy increases school dropout, reducing the labor force’s educational attainment and consequent economic productivity [18]. In recognition of the importance of safeguarding individuals’ reproductive health rights, advancing the cause of gender empowerment, and capturing beneficial economic development effects, there is long-standing global recognition of the importance of meeting AYA health needs [19,20,21,22]. In order to meet the health needs of AYAs and to make progress towards global family planning targets, ongoing efforts must be made to increase adolescent and youth sexual and reproductive health (AYSRH) service availability and quality.

Due to their level of cognitive and social development and their standing within their societies, AYAs have SRH service needs that are different from those of older adults [9]. AYAs oftentimes have few financial resources available with which to obtain SRH services. They also frequently experience barriers in seeking these services due to negative parental views and restrictive community and religious norms [23]. These normative barriers can be particularly severe for unmarried AYAs [24,25]. Their relative inexperience with reproductive health means that AYAs tend to require more time, counseling, and guidance on available SRH service options during consultations with health care providers [26]. In the specific case of adolescents, good counselling does not provide information alone, but also considers the developmental requirements of individuals in this age range. While adolescents increasingly demand autonomy in their decision making in order to affirm their own agency, they also still rely on the guidance of trusted adults when making important decisions [26]. Taken together, these factors can make AYAs a challenging group to care for. However, the rapid social, cognitive, and physical changes that mark this period of life makes adolescence and early adulthood a developmentally sensitive period during which, given sufficient investment in time and resources, individuals’ life-long health trajectories can be positively altered [27]. Providing young people access to youth friendly SRH services is a critical investment for any country to make. Doing so can ensure the rights and health of young people and their offspring and can serve to accelerate economic development in LMICs [9,28].

High levels of unmet need for SRH services, combined with the evident benefits to reproductive justice, gender empowerment, and economic development that these services provide, create a powerful argument for improving the quality of AYSRH services. In order to make these improvements, however, we must be able to define and measure quality unique to AYAs. The World Health Organization’s (WHO’s) quality of care framework for adolescent-friendly services characterizes adolescent-friendly health services as those that are accessible, acceptable, equitable, appropriate, and effective [29]. While not significantly different from quality of care frameworks for other ages, this and similar frameworks form the foundation to quality standards and assessment tools designed to inform improvements in health service provision to young people [30,31]. Efforts to assess and improve health service quality for AYAs have shown success in a number of LMICs [32].

In a review of indicators of youth-friendly health care, researchers found that there remains debate over the most important dimensions of AYSRH service quality and that many indicators reflect basic health care quality standards that are not specific to AYSRH [9]. More recently, during an assessment of FP2020’s contributions to advancing rights-based family planning, family planning specialists noted that there remains work to be done in building out indicators that facilitate program measurement, especially at the subnational level [33]. Indeed, while FP2030 has debuted a comprehensive results framework, few of the proposed indicators offer immediately actionable information to clinic- or district-level program managers [34].

### 1.2. Context

In order to capture the opinions of young people and service providers on what aspects of AYSRH quality of care serve as the most salient facilitators and barriers to seeking SRH services for AYAs, the authors conducted a contextual exploration of the subject by convening a series of focus group discussions (FGDs) with youth leaders and AYSRH program managers and experts. During FGDs, youth leaders from Vietnam, Nepal, Lebanon, Sudan, Ethiopia, Zimbabwe, and Poland overwhelmingly confirmed that the greatest barriers to obtaining high quality SRH counseling and services were poor privacy and confidentiality practices and provider bias. From the service delivery side, service delivery providers from numerous international health service organizations noted the important role that effective monitoring plays in quality improvement but expressed dissatisfaction with existing quality/youth friendliness measurement tools. Service providers mostly referred to WHO AYSRH assessments when suggesting that current methods are too complicated, do not measure the right dimensions of quality, and that data are frequently not useful for quality improvement. These experts expressed a need for standardized metrics that, while validated against recognized outcome measures of service adoption and client satisfaction, favored simplicity and brevity over comprehensiveness.

### 1.3. Goals

In order to meet this demand for quality improvement-oriented metrics in AYSRH service delivery, it is first necessary to explore the constructs that youth FGD participants noted as being the most salient impediments to high-quality SRH services for AYAs in their countries. In the context of AYSRH service delivery, privacy is “an individual’s ability to control disclosure or personal information, formulation and disclosure of beliefs and feelings, contact with others in social settings, and unwanted observation of body or intrusion of personal space.” [35] Confidentiality is defined as “an agreement between adolescent and provider that information discussed during or after the encounter will not be shared with other parties without the explicit permission of the patient.” [36] Provider bias refers to “attitudes and subsequent behaviors by providers that unnecessarily restrict client access and choice, often related to either client and/or contraceptive method characteristics.” [37] The purpose of this narrative review is to answer two research questions:How do poor privacy and confidentiality practices and provider bias limit the quality of AYSRH services and what facilitators are thought to mitigate these barriers?What assessment methods show the most promise for use in a standard performance monitoring measure of privacy, confidentiality, and provider bias is AYSRH quality of care?

## 2. Methods

A narrative review of the peer-reviewed and grey literature was conducted to answer the two research questions. Narrative reviews are appropriate for addressing research topics in which the existing body of literature is diverse in terms of methods, study settings, sample characteristics, and outcomes [38]. Databased peer-reviewed literature was searched through PubMed, Embase, and Google Scholar. Grey literature searches were performed on the online public report repositories of WHO, USAID Development Experience Clearinghouse, EngenderHealth, FP2020, The Challenge Initiative, Population Council, MEASURE Evaluation, International Planned Parenthood Federation, MSI Reproductive Choices, Population Services International, Adolescents 360, The YP Foundation, FHI360, CARE, Pathfinder International, and IntraHealth. Bibliographies of included literature were searched to identify additional relevant publications that may not have been captured by our search term strategy. No limitations were placed on study countries, but reports emanating from LMICs were given additional attention. Table 1 describes the review’s inclusion criteria.

For the database strategy, a list of terms describing the major constructs of privacy, confidentiality, and provider bias were compiled. To this list was included terms related to concepts of SRH service provision. Lastly, terms describing the concepts of adolescents and young people were added to search protocols. The Boolean operation “AND” was used to combine different concepts and “OR” was used to capture articles containing at least one of the individual terms used to define each concept. Database index terms, such as MesH terms, were included when available. Gray literature searches were adapted to meet the capabilities of each organization’s website. Table 2 provides key search terms for each concept.

Database search results were exported to Covidence, a primary screening and data extraction program. Duplicates were removed by the program as search results were collated. One author (AC) conducted title and abstract reviews of all articles using the described inclusion criteria. The author evaluated titles and abstracts of all studies and retained those studies that were believed to be relevant. Potentially relevant studies were then retrieved, and full texts were reviewed by the author. From studies found to be relevant to the review, data that helped to answer one of the review’s overarching questions were extracted. Extracted data included authors, publication year, study locations, sample characteristics, effects of poor privacy and confidentiality practices and provider bias, and measurement methods employed. This extracted data was then analyzed by the larger author group (AC, AS, DM, and NC) and findings were synthesized.

## 3. Results

A total of 3254 articles were retrieved from PubMed, 1870 from Embase, and the first 150 entries from Google Scholar. Of the 5274 studies retrieved from the three databases, 730 were found to be duplicates and were immediately removed. An additional 4342 articles were removed during title and abstract reviews. During the full-text review, 107 of the 202 retrieved articles were excluded for not meeting inclusion criteria. An additional 16 documents were identified as relevant while searching organizations’ online document repositories. Figure 1 details article selection.

Of the 111 documents included in the review, 102 were studies or evaluations taking place in 42 countries. Countries were found in Africa, Asia, Australia, Europe, and North America. No studies identified for this review were conducted in South America. Figure 2 illustrates the geographic distribution of study and evaluation locations. AYAs were the primary participant sample in 41 of these 102 studies. Table 3 and Table 4 provide details on the articles, reports, and assessment tools identified during this review.

### 3.1. How Do Poor Privacy and Confidentiality Practices and Provider Bias Limit the Quality of AYSRH Services and What Facilitators Are Thought to Mitigate These Barriers?

**Poor privacy and confidentiality practices**. The development of increased privacy needs and communities’ enforcement of gender and social norms occur in the early years of adolescence [148]. The emergence of these factors mean that privacy and confidentiality concerns are most common at younger ages of adolescence and are associated with a lower likelihood of receiving SRH services than adult clients [63,71]. Social norms that stigmatize adolescent sexual development can lead to feelings of shame and embarrassment and cause adolescents to fear that their family or other community members will learn of their SRH seeking behaviors [48,50,63,82,99,116,132]. In order to meet developmental needs and sidestep potential social sanctions, AYSRH-friendly services prioritize client privacy and confidentiality [149]. A lack of appropriate levels of privacy and confidentiality is a marker of poor health service quality and dissuades AYAs from future health seeking behaviors [41,61,69,70,73,78,97,120,141].

A number of factors are found throughout the literature that have been implicated in reducing privacy and confidentiality in AYSRH. Health care providers’ own lax attitudes towards client confidentiality frequently undermine AYAs’ belief that their health information will remain private. AYA participants in numerous studies have reported either fearing or previously experiencing breaches in their confidentiality by health care providers [41,47,50,61,66,71,73,78,80,85,88,90,93,96,98,102,121,125,130]. Violations can occur when providers discuss clients’ health information in public areas or disclose clients’ information to family or other community members without prior consent. Poor health facility layouts and operations can also reduce AYA clients’ sense of privacy. Points of service without private waiting or consultation rooms or toilets; frequent interruptions during consultations; and poor health records management have all been cited by AYAs as point of service features that threaten client privacy and confidentiality [43,78,80,85,88,95,98,106,110,117,129,130].

In order to avoid threats to their privacy, AYAs in some studies reported preferring to visit private clinics, traditional healers, and pharmacies. These alternatives to public health clinics frequently have the reputation of being more confidential [44,46,58,99,110,121,132]. So important is service privacy to young clients that the authors of one study noted that rather than requesting condoms from public providers, Ugandan youth would pay out-of-pocket from pharmacies using coded terminology, all to avoid disclosure [88]. These alternative service points, however, tend to offer less client counseling and tend to be more expensive than publicly funded options.

**Provider bias.** The effects of provider bias against AYAs are substantial. In numerous studies, provider attitudes were one of the most important factors in AYAs’ consideration of SRH service quality [53,65,92,107,121,136]. Provider bias frequently manifests as limitations based on clients’ age and parity. Providers oftentimes set minimum client age and number of children requirements for products such as long-acting reversible contraceptives (LARCs), contraceptive injections, oral contraceptives (OC) and emergency contraception (EC) [40,42,52,55,56,64,72,83,104,110,111,122,123,124,126,127,128,134]. Providers also report withholding family planning methods for nulliparous women or even those who were deemed to have too few children [89,97,102,103]. Similarly, providers sometimes impose limitations based on marital status or partner approval. Provider-imposed limitations based on marital status or spousal approval are a common provider barrier encountered by AYA women seeking LARCs, OC, and EC [62,104,118,124,127,128,134,139]. Establishing age, parity, and marriage requirements outside of those outlined by recognized family planning guidelines was justified by many providers as necessary based on misconceptions about contraception contraindications or adverse effects, or in order to discourage adolescent promiscuity [65,67,86,100,103,138].

Provider bias has also been linked to method bias. Insufficient or poor quality counseling of AYA clients on available contraception options was mentioned in numerous publications as an effect of provider bias [54,55,57,68,84,89,104,110,112,139]. When AYA clients have been able to receive SRH services, many reported not receiving their preferred contraceptive method [79,112,118,128]. Providers’ method bias has been linked to providers’ inexperience and resulting discomfort in administering certain contraceptive methods, misunderstandings regarding the appropriateness of those methods for AYAs, or paternalistic attitudes and disregard for clients’ opinions [67,79,91,118,123,131,137,139,140]. Not receiving one’s desired contraceptive method or receiving poor quality counseling can lead to method dissatisfaction, a key risk factor for contraceptive discontinuation [150,151].

Lastly, provider bias and negative provider attitudes can lead to providers yelling at, scolding, chastising, or otherwise stigmatizing AYA clients for seeking SRH services. Stigmatization of AYAs features prominently in the scholarship as a barrier to AYA health seeking. Stigmatization is a powerful force that discourages clients from disclosing important health history and from future health seeking behaviors. AYAs described provider stigmatization as a major barrier to seeking out HIV and sexually transmitted infection (STI) testing, treatment, and counseling; family planning products; and abortion and post-abortion care services [64,68,69,80,81,104,108,124,125,126,127,128,129]. The influence of providers’ attitudes on AYAs’ perceptions of available health services and their decision to seek care cannot be understated. In Botswana, provider attitudes were the greatest predictor of AYAs’ perceptions of health facility services, while a survey of migrant women working as beer promoters in Thailand, Laos, Vietnam, and Cambodia concluded that a large majority of respondents chose health care institutions based on the friendliness of its providers [92,136].

**Strategies to improve privacy and confidentiality.** Prioritizing AYA clients’ privacy and confidentiality is fundamental to earning AYAs’ trust. Ensuring clients’ privacy has been associated with increases in SRH service discussion and uptake [49,51,63,113,115]. Additionally, greater levels of facility privacy have been shown to be protective against contraceptive method discontinuation [59]. When asked, youth participants in several studies did not prioritize standalone youth services or youth centers [66,75,85]. This suggests that health systems and facilities in very austere conditions may be able to improve AYA privacy and confidentiality without investing in operating dedicated youth service programs.

The literature contains several well documented strategies to improve the privacy and confidentiality of AYSRH. Health systems offering high quality AYSRH services recognize the importance of AYA client privacy and confidentiality. This involves organizing clinics in ways that facilitate privacy while waiting and during clinic visits, and encouraging providers to assure their young clients that their personal information will remain confidential [50,60,69,70,78,82,88,90,101,108,121,133]. Making certain services and products easily accessible can also facilitate service use by reducing feelings of client embarrassment [50,60,82,88,121]. A strategy such as this might include having sexual health literature available in clinic waiting rooms and pharmacy front areas so that AYA clients can take it as they leave the point of service without having to make an explicit request for such materials [101]. Additionally, clinics and pharmacies can make condoms available, for sale or for free, in discreet packaging without the need for an appointment or consultation [50,60,82,120,121].

Finally, points of service can consider extending hours in order to improve AYA privacy and confidentiality. Offering extended or dedicated service hours to AYAs is not only a convenience but is also another way to facilitate AYA privacy. Extending operation hours allows AYAs to avoid clinic hours during which other patients are present, such as those seeking maternal and child health services, and could better accommodate youth school and working hours [41,43,45,46,61,78,87,101,117]. Finally, practicing other aspects of AYA-friendly service provision, such as freely providing age-appropriate educational materials, encouraging AYA feedback, and engaging in community outreach, can all indirectly improve quality of care and AYA service uptake [101,115].

**Strategies to mitigate provider bias.** Providers’ attitudes toward providing appropriate AYSRH services are not intransigent. In-service education on relevant AYA contraception guidelines or training designed to increase provider competence in providing SRH services, such as LARC implantation, can improve provider attitudes towards the AYAs who seek them [42,72,124,131,140]. However, most provider bias is not due to gaps in knowledge or technical expertise, and so guideline clarification or additional training is often insufficient in addressing it [42]. More often, provider biases against AYAs are deeply rooted in social norms and community attitudes related to the agency of women and young people, as well as the importance of female abstinence before marriage and demonstrating fertility soon afterwards [26,67,124,128,135]. 

Two strategies that have shown promise in countering restrictive social and gender norms include values-reflection and provider mentorship. Health systems or social franchise networks can consider implementing values-reflective trainings or experiences that ask providers to view circumstances through their clients’ perspective or to reflect on personal values. Values-reflective interventions are one approach to improve client-provider communication and to counter restrictive social and gender norm drivers of provider bias towards AYSRH [76,152]. Alternatively, health facilities can recruit providers who have demonstrated providing high-quality care to AYAs as mentors or role models for other providers. As social norm theory suggests, respected colleagues have the power to shape social norms around the acceptability of delivering high-quality AYSRH services [153]. These “positive deviant” providers can be influential sources of information and could promote expansion of the range of contraceptive methods their colleagues provide to AYA clients [37,67].

### 3.2. What Assessment Methods Show Promise for Use in a Standard Performance Monitoring Measure of Privacy, Confidentiality, and Provider Bias in AYSRH Quality of Care?

Twenty-five of the documents retrieved for this review assessed the quality of AYSRH service provision for individuals or clusters of service delivery points. Studies relied on one or more methods to explore privacy, confidentiality, and provider bias: client interviews; provider interviews; simulated clients; provider observation; facility audit; and health records audits. Table 5 provides the frequency with which each method was used.

Quantitative client interviews were conducted in eight studies using structured surveys, while three studies relied on qualitative methods, such as individual interviews (IDIs) or FGDs, to assess clients’ opinions on AYSRH service quality [39,40,68,77,97,99,112,117,129]. Provider interviews were conducted using similar quantitative and qualitative methods [39,40,55,67,68,75,99,104,106,112,117,122,131]. While assessment methods might be comparable between studies, there was significant variation in how they were applied to explore the concepts of poor privacy and confidentiality practices and provider bias. For instance, when examining provider bias in post abortion care (PAC) services, Evens et al. (2014) used quantitative client surveys to explore perceived levels of provider respect, compassion, judgement, and negative attitudes towards young clients [68]. In contrast, Sovd et al. (2006) assessed provider bias by surveying young clients on whether providers listened to them carefully, gave opportunities to ask questions, did not limit the number of questions clients could ask, and treated them in the manner in which they wanted to be treated [129]. Perhaps similar in some respects, these studies operationalize provider bias in meaningfully different ways.

Two studies used vignettes to explore providers’ potential biases [64]. Vignettes are short descriptions of a person, object, or situation that are designed to represent a combination of characteristics of interest [154]. In measuring provider bias, vignettes have the potential to allow evaluators to alter situations in order to identify client characteristics that elicit biased attitudes. Simulated clients were employed on a number of occasions, with a combination of quantitative and qualitative debriefings taking place after clinical encounters [67,75,92,95,98,106,109,110,116,128]. Sieverding et al. (2018) used two vignettes to explore the scope of family planning services providers would offer to an unmarried, childless 18-year-old woman who is in school and wishes to avoid getting pregnant, versus those they would offer to a married 28-year-old with two children who also wishes to avoid getting pregnant again in the near future [128]. Dieci et al. (2021) explored the effects of the same client-level factors on provider bias (i.e., client age, marital status, and parity), but employed different combinations of these factors to create 18 unique client profiles that serve to isolate the client-level factors that acted as the principle drivers of provider bias [64]. The increased explanatory power offered by the latter study is due, in part, to a significantly larger sample size (790 vs. 52).

Health care provider observation was used on two occasions [40,117]. Provider observation allows for assessment of provider knowledge and clinical performance, but can be challenging and costly to conduct due to confidentiality concerns [64]. Additionally, the presence of an observer during clinical encounters may alter the behaviors of the provider, leading to a Hawthorne effect bias. Facility audits were conducted in several cases to investigative the availability of supplies, counseling literature, and relevant national or international SRH guidelines for working with AYAs [39,40,55,68,117]. Facility audits did not include evaluations of the clinics’ waiting or counseling areas [99,104]. Finally, two studies evaluated AYSRH service quality by using health records to examine trends in AYA clinic visits and choice in contraceptive methods. These research methods are helpful to enumerate a single point in time quality but are not easily implemented for routine assessment of quality AYSRH delivery. 

Over the last two decades, the World Health Organization, its regional offices, and a number of global non-profit organizations, have created frameworks and assessment tools designed to measure AYSRH service quality [30,31,142,143,144,145,146,147]. Assessment instruments, such as the WHO South-East Asia Regional Office’s “Adolescent Friendly Health Services Supervisory/Self-Assessment Checklist: User’s Guide”, provide health service supervisors with a means by which to assess whether services meet the established standards. Several studies reported adopting one of these or similar evaluation tools to conceptualize and assess the multiple dimensions of AYSRH quality [39,40,92,95,97,99,106,109,116]. However, these studies, that evaluated the quality of AYSRH service delivery, did not correlate quality domains or individual indicators to measures of AYA satisfaction, contraceptive method adoption and maintenance, or other similar client-level outcomes of interest.

## 4. Discussion

SRH services and products are some fundamental offerings of any primary care health system and must address the needs of the entire population of individuals of reproductive age. Ensuring AYAs have access to high quality SRH services is an important step in helping them to achieve sexual and reproductive health and well-being and is essential in meeting the global community’s broader SRH targets [21,22,155]. Impressive strides have been made in increasing SRH coverage in many LMICs and large-scale interventions have proven that both supply- and demand-side interventions can increase the adoption of modern FP methods by both ensuring an adequate supply of products, and by increasing public awareness of their health and economic benefits [156,157,158,159]. However, ample SRH service coverage that meets demand, and increased awareness of FP benefits, may be inadequate if poor privacy and confidentiality practices and provider bias remain unaddressed barriers to AYSRH provision. 

As the results of this review suggest, issues of privacy, confidentiality, and provider bias are present throughout most of the country; these also include cultural and health system contexts and a significant degradation in AYSRH service quality. While not inclusive of all the factors that drive care quality, these concepts appear to disproportionately influence clients’ perceptions of available AYSRH services. Poor privacy and confidentiality practices and provider bias all have the potential to dissuade AYAs from seeking care in the first place. Those that do risk facing a health system that is ill-prepared to provide care in a developmentally appropriate manner that is respectful of young people’s autonomy and sexual and reproductive health rights. Fortunately, as previously noted, there are tactics that health systems and service delivery points can adopt to improve privacy and confidentiality and reduce provider bias. Additionally, implementation guides designed to support health facilities to address issues around privacy, confidentiality, and provider bias in AYSRH service provision are available to organizations seeking to develop solutions specific to their needs [160]. In order to establish whether quality improvements have the desired effect on AYSRH quality, it is necessary to regularly monitor service delivery quality. 

Regular service monitoring is an important element of any performance improvement process [75,117]. Studies and evaluations included in this review employed numerous assessment methods to evaluate the quality of AYSRH services. Client- and provider-centered qualitative methods and surveys, vignettes, simulated clients, direct observation, facility audits, and health record reviews are all valid assessment methods that each have their own combination of strengths and weaknesses [161]. Evaluators of AYSRH programs must consider which components of service quality are being judged before selecting a method. For instance, the impact of providers’ potentially biased attitudes and behaviors towards AYA clients is likely better assessed through simulated clients than through provider survey methods. Alternatively, privacy and confidentiality could be effectively assessed through AYA-led facility audits or client surveys. 

As this review has outlined, a number of AYSRH-friendly guidelines and assessments exist to aid program supervisors in determining whether services can be considered AYA-friendly or are meeting established standards [30,31,142,143,144,145,146,147]. Being certified as AYA-friendly can be an important signal to the youth that a center will treat them with dignity and respect, but this certification does not replace regular performance monitoring to ensure a consistent high quality service delivery. At present, most assessments of AYSRH service quality cover numerous domains that give equal importance to all domains of AYSRH quality. Additionally, existing indicators lack standardization and do not sufficiently demonstrate how youth-friendly service provision and utilization drive improved health outcomes. Research is needed that better defines which point of service factors have the greatest influence over client satisfaction and health outcomes [9]. Standard, easily operationalized indicators that have been validated against relevant AYSRH client satisfaction and health outcomes are a prerequisite to quality improvement and would play an important role in creating friendlier and more frequently utilized AYSRH service points [162,163,164]. 

In order to create easy-to-deploy, contextually relevant service quality monitoring assessments there remains a need to differentiate between those service delivery points and provider characteristics that may have the greatest influence on perceived QOC and young people’s service use [9]. To create this type of a measure, rigorous studies must be conducted to refine the list of given indicators by retaining only those most strongly correlated to AYA SRH outcomes of interest, including privacy, confidentiality, and provider bias. This important step will help us to measure the components of AYSRH that matter most to AYAs and assess the impact of service improvement efforts on AYA SRH health outcomes.

## 5. Limitations

Limitations of this review should be considered. The purpose of this review was to answer a series of AYSRH service quality questions. As such, a narrative rather than systematic review method was selected. However, narrative reviews cannot offer the same degree of reproducibility, critical appraisal, and evidence synthesis as systemic reviews. Second, the review’s wide geographic scope means that key concepts related to poor privacy and confidentiality and provider bias needed to be abstracted in order to be more broadly relevant. This should be remembered when considering a specific country or region. The ways in which poor privacy and confidentiality or provider bias operationalize themselves and how AYA respond to these challenges are context specific. 

## 6. Conclusions

As the results of this review suggest, issues of privacy, confidentiality, and provider bias are present throughout most country, cultural, and health system contexts and significantly degrade AYSRH service quality. While not inclusive of all the factors that drive care quality, these concepts appear to disproportionately influence clients’ perceptions of available AYSRH services. Fortunately, there are tactics that health systems and service delivery points can adopt to improve privacy and confidentiality and to reduce provider bias. In order to establish whether quality improvements have the desired effect on AYSRH quality, it is necessary to regularly monitor service delivery quality. 

Although AYSRH-friendly guidelines and assessments exist, they are not a replacement for routine performance monitoring to ensure delivery of high-quality services. Existing indicators of AYSRH quality services lack standardization and do not sufficiently demonstrate how youth-friendly service provision and utilization drive improved health outcomes. Research is needed that better defines which point of service factors have the greatest influence over client satisfaction and health outcomes [9]. This important step will help us to measure the components of AYSRH that matter most to AYAs and assess the impact of service improvement efforts on AYA SRH health outcomes.

## Figures and Tables

**Figure 1 ijerph-19-06576-f001:**
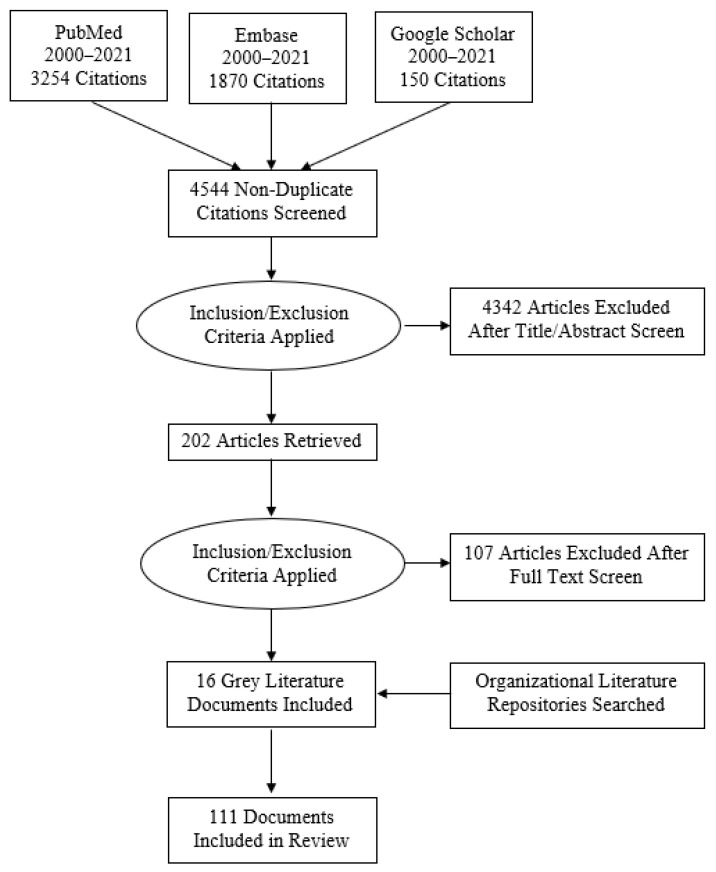
Literature selection diagram.

**Figure 2 ijerph-19-06576-f002:**
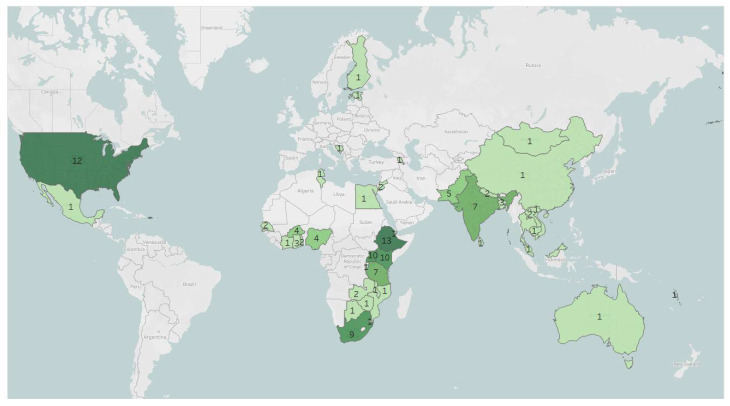
Study countries, including number of studies conducted within each country. (Note: Color shading corresponds to the number of studies conducted within a given country, with darker shading corresponding to a greater number of studies. Some studies were conducted in multiple countries).

**Table 1 ijerph-19-06576-t001:** Inclusion criteria.

Time Frame	2000–2021
Study design	Quantitative, qualitative, or mixed-methods research
Study population	Adolescents and young adults aged 10–25
Geographic scope	Global, with particular interest paid to studies from low- and middle-income countries
Outcomes	(1) Descriptions of how poor privacy and confidentiality practices and provider bias served as barriers to AYAs seeking SRH services *. OR (2) Descriptions of point of service factors that mitigated the risk of poor privacy and confidentiality practices or provider bias and facilitated AYA service seeking behaviors. OR (3) Descriptions of assessment methods used in evaluating privacy, confidentiality, and provider bias in AYSRH points of services.
Language	English or French
Document type	Peer-reviewed journal articles, project evaluations, reports, or presentations

* SRH services: modern contraceptive methods; abortion and post-abortion care (PAC) services; and HIV and sexually transmitted infection testing and counseling.

**Table 2 ijerph-19-06576-t002:** Key search terms used to detect database literature (PubMed, Embase, Google Scholar) *.

Privacy, Confidentiality, and Provider Bias	Sexual and Reproductive Health		Adolescents and Young Adults
Privacy Confidentiality Provider bias Provider attitudes Staff attitudes Age discrimination Minimum age restrictions Provider barriers Medical barriers Eligibility restrictions	Reproductive health services Family planning Contraception Sexually transmitted infection HIV testing Birth control Intrauterine device Long-acting reversible contraception	Emergency contraception Injectable Abortion Pregnancy Patch Implant	Youth Young adult Adolescent Teenager Early adolescent Pre-teen Pre-adolescence

* Index terms used when available.

**Table 3 ijerph-19-06576-t003:** Research studies and evaluations included in review.

Authors	Year	Countries	Research and Evaluation Reports	Sample
			SRH Services of Interest	
Abdel-Tawab et al. [39]	2015	Egypt	Family Planning	SRH clinics (*n* = 40)
Abebe et al. [40]	2012	Ethiopia	Family Planning	Health facilities (*n* = 113), health care providers (*n* = 182), and clients (*n* = 457)
Agampodi et al. [41]	2008	Sri Lanka	STI/HIV prevention and treatment Family Planning	Adolescents aged 17–19 (FGD *n* = 4)
Agha et al. [42]	2011	Pakistan	Family Planning	Clinical providers (*n* = 566)
Akatukwasa et al. [43]	2019	Uganda	STI/HIV prevention and treatment Family Planning	Young people (*n* = 48), health care providers (*n* = 63), and key informants (*n* = 11)
Anand and Sinha [44]	2010	India	STI/HIV prevention and treatment Family Planning	Women aged 15–39 (*n* = 7785)
Ansha et al. [45]	2017	Ethiopia	STI/HIV prevention and treatment Family Planning	Adolescents aged 15–19 (*n* = 402) and community stakeholders (FGDs *n* = 4, and IDIs *n* = 10))
Ayehu et al. [46]	2016	Ethiopia	STI/HIV prevention and treatment Family Planning	AYAs aged 10–24 (*n* = 746)
Berhane et al. [47]	2005	Ethiopia	STI/HIV prevention and treatment Family Planning	AYAs aged 10–24 (*n* = 2647)
Biddlecom et al. [48]	2007	Burkina Faso Ghana Malawi Uganda	STI/HIV prevention and treatment Family Planning	AYAs aged 12–19 (3457)
Binu et al. [49]	2018	Ethiopia	STI/HIV prevention and treatment Family Planning	AYAs aged 15–24 (*n* = 739)
Birhanu et al. [50]	2018	Ethiopia	Family Planning	Adolescents aged 13–18 (*n* = 1262) and FGDs (*n* = 8) with students, health care providers, and parents.
Bostick et al. [51]	2020	United States	Family Planning	Sexually active adolescent girls (*n* = 730)
Bryce et al. [52]	2016	Uganda	Family Planning	Client women aged 18–49 (*n* = 92)
Burke et al. [53]	2017	Senegal	Family Planning	Young people with disabilities (*n* = 50 IDIs and 17 FGDs)
Calabretto et al. [54]	2005	Australia	Emergency contraception	Young women who had used emergency contraception (*n* = 13)
Calhoun et al. [55]	2013	India	Family Planning	Healthcare providers (*n* = 1752) and pharmacist centers (*n* = 517)
Camber Collective [56]	2018	Burkina Faso Pakistan Tanzania	Family Planning	Health care providers in public and social franchise points of services (*n* = 1050)
Capurchande et al. [57]	2016	Mozambique	Family Planning	AYAs aged 15–24 (*n* = 16 IDIs)
Cartwright et al. [58]	2019	40 countries	Family Planning	AYAs aged 18–35 (*n* = 207)
Chang et al. [59]	2020	Pakistan Uganda	STI/HIV prevention and treatment Family Planning	Women aged 15–49 (*n* = 1998)
Char et al. [60]	2011	India	Family Planning	Unmarried AYA men aged 17–22 in FGDs (*n* = 4) and survey (*n* = 316)
Cherie and Berhane [61]	2012	Ethiopia	STI testing and prevention	AYAs aged 15–24 (*n* = 3543 and FGD *n* = 8)
Collumbien et al. [62]	2011	India	Family Planning Abortion services	AYAs aged 15–24 (*n* = 6572) and service providers (*n* = 264)
Copen et al. [63]	2016	United States	Family Planning	Not provided
Dieci et al. [64]	2021	Burkina Faso Pakistan Tanzania	Family Planning	Health care providers (*n* = 790)
Dixit et al. [65]	2015	India	Emergency contraception	OBGYN FGDs (*n* = 3) and key informant interviews
Erulkar et al. [66]	2005	Kenya Zimbabwe	STI/HIV prevention and treatment Family Planning	AYAs aged 10–19 (*n* = 1883)
Esso et al. [67]	2017	Ivory Coast	Family Planning	Nurses and midwives (FGDs *n* = 15 and IDIs *n* = 15)
Evens et al. [68]	2014	Kenya	Abortion services	Client women aged 16–49 (*n* = 283)
Ezenwaka et al. [69]	2020	Nigeria	Family Planning	Policy makers, legislator, program managers, implementing partners, local nongovernment organizations, community leaders, health service providers and parents/caregivers of unmarried adolescents (IDIs = 81 and FGDs = 6).
Flaherty et al. [70]	2005	Uganda	Family Planning	Adolescents aged 14–20 (FGD *n* = 4)
Fuentes et al. [71]	2018	United States	Family Planning	AYAs aged 15–25 (*n* = 2325)
Gausman et al. [72]	2021	Jordan	Family Planning STI testing	Primary care physicians, midwives, and nurses (*n* = 510)
Gautam et al. [73]	2018	Nepal	Family Planning	AYAs aged 15–24 (IDIs *n* = 22)
Geary et al. [74]	2014	South Africa	STI testing HIV testing and counseling Family Planning	Clinic charge nurses (*n* = 7)
Geary et al. [75]	2015	South Africa	STI/HIV prevention and treatment Family Planning	Primary healthcare clinics (*n* = 15)
Geibel et al. [76]	2017	Bangladesh	HIV counseling & testing	Social franchise providers (*n* = 300) and key population AYAs aged 15–24 (*n* = 266 and 371)
Haller et al. [77]	2012	Bosnia and Herzegovina	STI/HIV prevention and treatment Family Planning	Young people aged 11–24 (*n* = 60)
Hayrumyan et al. [78]	2020	Armenia	Family Planning	Adolescents aged 18–19 (*n* = 17) and PHPs
Higgins et al. [79]	2016	United States	Family Planning	Young women (*n* = 50) with a history of intrauterine device or implant use aged 18–29.
Hokororo et al. [80]	2015	Tanzania	Family Planning Antenatal care STI/HIV testing	Pregnant women aged 15–20 (FGDs *n* = 9)
Jain et al. [81]	2020	India	Family Planning	AYAs aged 17–26 (*n* = 200)
Jonas et al. [82]	2020	South Africa	Family Planning HIV counseling & testing	AYA girls aged 15–24 (FGDs *n* = 19 & IDIs *n* = 57)
Judge et al. [83]	2011	Kenya Ethiopia	Emergency contraception	Emergency contraception providers (Kenya *n* = 523 and Ethiopia *n* = 121)
Kavanaugh et al. [84]	2013	United States	Family Planning	Facility administrative director IDIs (*n* = 20), facility staff FGDs (*n* = 6), and IDIs with clients (*n* = 48) aged 16–24.
Kennedy et al. [85]	2013	Vanuatu	Family Planning	Adolescents aged 15–19 (FGDs *n* = 66), policy makers, and service providers.
Khan et al. [86]	2014	India	Emergency contraception	Physicians providing emergency contraception (*n* = 83)
Kiapi-iwa and Hart [87]	2004	Uganda	STI testing HIV testing and counseling Family Planning	AYAs aged 10–20 (IDIs *n* = 11)
Kipp et al. [88]	2007	Uganda	STI testing HIV testing and counseling Family Planning	Youth-related NGO leaders (*n* = 4), nurses (*n* = 2), and youth leaders (*n* = 4)
Kohn et al. [89]	2012	United States	Family Planning	School-based health center staff and clinicians (*n* = 162)
Kyilleh et al. [90]	2018	Ghana	Family Planning	AYAs aged 10–19 (FGD *n* = 8) and stakeholder (IDIs *n* = 9)
Leroy-Melamed et al. [91]	2021	United States	Family Planning	Clinical providers (*n* = 78)
Lesedi et al. [92]	2011	Botswana	Family Planning	Youth 15–29 years (*n* = 110)
Lince-Deroche et al. [93]	2015	South Africa	Family Planning	AYA women aged 18–24 (IDI *n* = 90)
MacPhail et al. [94]	2008	South Africa	HIV counseling and testing	AYA (*n* = 24) and parent (*n* = 12) FGDs
Mathews et al. [95]	2009	South Africa	STI/HIV prevention and treatment	Mystery client visits to 93 clinics
Matseke et al. [96]	2016	South Africa	HIV counseling and testing	HIV counseling and testing clinic clients (*n* = 498)
Mayeye et al. [97]	2010	South Africa	STI/HIV prevention and treatment Family Planning	Adolescents aged 16–19 visiting 11 public health clinics
Mchome et al. [98]	2015	Tanzania	STI/HIV prevention and treatment Family Planning	Forty-eight visits to health facilities (*n* = 33)
Mmari and Magnani [99]	2003	Zambia	STI/HIV prevention and treatment Family Planning	Public clinics (*n* = 10)
Mngadi et al. [100]	2008	Swaziland	Family Planning STI testing HIV counseling and testing Abortion services	Health care providers (*n* = 56)
Moise et al. [101]	2017	Burundi	STI/HIV prevention and treatment Family Planning	Health facility survey (*n* = 892) and total AYAs aged 10–24 visiting facilities over last seven days (*n* = 24,232)
Molla et al. [102]	2009	Ethiopia	STI testing and treatment	Sexually active AYAs aged 15–24 (*n* = 3743) and health care providers (IDIs *n* = 10)
Morgan et al. [103]	2019	United States	Family Planning	Providers (*n* = 2056)
Mugore et al. [104]	2019	Togo	Abortion services	Expert opinion
Mulaudzi et al. [105]	2018	South Africa	HIV counseling and testing	HIV providers and counselors (*n* = 2 FGDs & *n* = 19 IDIs)
Munea et al. [106]	2020	Ethiopia	STI/HIV prevention and treatment Family Planning	Health facilities (*n* = 18), their health care providers (*n* = 36), and key informants (*n* = 8)
Murithi et al. [107]	2020	Burkina Faso Pakistan Tanzania	Family Planning	n/a
Mutea et al. [108]	2020	Kenya	Family Planning	Adolescents, community representatives, teachers, health care providers, and county leaders
Nalwadda et al. [109]	2011	Uganda	Family Planning	Simulated client visits (*n* = 128) by AYAs aged 15–24
Nalwadda et al. [110]	2016	Uganda	Family Planning	Five female and two male simulated clients (SCs) interacted with providers (*n* = 128) at public, private not-for-profit, and private for-profit health facilities.
Newport et al. [111]	2019	Nigeria Ethiopia Tanzania	Family Planning	Key informant interviews (*n* = 318) and FGDs (*n* = 64)
Ontiri et al. [112]	2019	Kenya	Family Planning	Female clients seeking FP services (*n* = 423) aged 15 and over (mean 28.3 ± 7.3 y) and healthcare providers (*n* = 12).
O’Sullivan et al. [113]	2010	United States	Family Planning	Primary care physicians (*n* = 21)
Otsin et al. [114]	2021	Ghana	Abortion services	Women who had experienced abortion complications (*n* = 24), and formal (*n* = 10) and informal abortion providers (*n* = 13)
Part et al. [115]	2016	Estonia	Family Planning	Women aged 16–24 (*n* = 868)
Pastrana-Sámano et al. [116]	2020	Mexico	STI testing and treatment	Adolescent-friendly clinics (*n* = 11)
Pathfinder International [117]	2019	Bangladesh	Family Planning	Health facilities (*n* = 69)
Pleasants et al. [118]	2019	Togo	Family Planning	Healthcare providers (*n* = 45) and female clients (*n* = 619) with mean age of 30 ± 6.5 (SD)
Raifman et al. [119]	2018	Tunisia	Abortion services	Abortion providers (*n* = 23)
Regmi et al. [120]	2010	Nepal	Family Planning	AYAs aged 18–22 (FGDs *n* = 10 & IDIs *n* = 31)
Robert et al. [121]	2020	Kenya	STI testing HIV testing & counseling Family Planning	Adolescents aged 10–19 (*n* = 9 FGDs and 18 IDIs); boys reporting engagement in same sex relationships, girls engaged in sex work, and males and females engaged in IVDU
Sannisto and Kosunen [122]	2009	Finland	Family Planning	Administrators and clinician (*n* = 208) at 63 Health center organizations
Scholl et al. [123]	2004	Ethiopia	HIV testing and counseling Family Planning	Key informant interviews (*n* = 52)
Schwandt et al. [124]	2017	Nigeria	Family Planning	Doctors, nurse/midwives, and community health extension workers (CHEWs) (*n* = 1479); pharmacists (*n* = 415); and patent medical vendors (PMV) (*n* = 483).
Sedekia et al. [125]	2017	Tanzania	Family Planning	Community members of both sexes (FGDs *n* = 71) and community women (IDI *n* = 18)
Senlet et al. [126]	2016	Bangladesh	Family Planning	Health facilities (*n* = 14) in four districts
Sidze et al. [127]	2014	Senegal	Family Planning	Young women (*n* = 2577) aged 15–29 and healthcare providers (*n* = 637).
Sieverding et al. [128]	2018	Nigeria	Family Planning	Two separate simulated client visits to private FP providers (*n* = 55; pharmacies, PPMVs, and licensed community health workers) and follow-up interviews with providers (*n* = 52)
Sovd et al. [129]	2006	Mongolia	Family Planning	Adolescents aged 10–19 (*n* = 1301)
Tangmunkongvorakul et al. [130]	2012	Thailand	Family Planning	Unmarried AYAs aged 17–20 (*n* = 1745)
Thompson et al. [131]	2018	United States	Family Planning	Clinic providers (*n* = 576)
Thongmixay et al. [132]	2019	Laos	Family Planning	AYAs aged 15–25 (IDIs *n* = 29) and health providers (IDIs *n* = 7)
Thrall et al. [133]	2000	United States	Pelvic exam	Adolescents in 9th and 12th grade (*n* = 1715)
Tumlinson et al. [134]	2015	Kenya	Family Planning	FP providers (*n* = 676)
Warenius et al. [135]	2006	Kenya Zambia	Family Planning STI testing HIV counseling and testing Abortion services	Nurse mid-wives (*n* = 820)
Webber et al. [136]	2012	Cambodia Laos Thailand Vietnam	Family Planning Pre-natal care Abortion	Beer promotion women (*n* = 22 FGDs and *n* = 390 survey respondents w/mean age of 24.2 yrs) and key informants
Wesson et al. [137]	2008	Kenya	Family Planning	FP providers and CBDs (*n* = 522)
Women’s Refugee Commission and UNHCR [138]	2011	Djibouti Kenya Uganda Jordan Malaysia	Family Planning	Not provided
Yirgu et al. [139]	2020	Ethiopia	Family Planning Ante-natal care	Client FGDs (*n* = 10) and IDIs (*n* = 30) with women 15–49 and men 18+
Zapata et al. [140]	2019	United States	Family Planning	FP providers in OBGYN, family medicine, adolescent medicine, and Title X clinics (*n* = 3445)
Zhang et al. [141]	2004	China	HIV prevention services	AYAs aged 15–24 (*n* = 1227)

**Table 4 ijerph-19-06576-t004:** AYSRH-friendly assessment tools included in review.

Assessment Tools		
Authors	Year	Assessment Title
IPPF [142]	2008	Provide: Strengthening Youth Friendly Services
IPPF [143]	2015	Provide: Self-Assessment Tool for Youth Services
Hainsworth et al. [144]	2004	Certification Tool for Youth Friendly Services
Pathfinder International [145]	2002	Clinic Assessment of Youth Friendly Services: A Tool for Assessing and Improving Reproductive Health Services for Youth
PSI [146]	2014	Making Your health Services Youth-Friendly: A Guide for Program Planners and Implementers
WHO and UNAIDS [31]	2015	Global Standards for Quality Health-Care Services for Adolescents: A Guide to Implement a Standards-Driven Approach to Improve the Quality of Health Care Services for Adolescents
WHO [30]	2009	Quality Assessment Guidebook: A Guide to Assessing Health Services for Adolescent Clients
WHO-SEARO [147]	2011	Adolescent Friendly Health Services Supervisory/Self-Assessment Checklist: User’s Guide

**Table 5 ijerph-19-06576-t005:** Point of service assessment methods.

Clients	Quantitative—surveys	8
	Qualitative—IDIs and FGDs	3
Providers	Quantitative—surveys	9
	Qualitative—IDIs and FGDs	9
	Vignettes	2
Simulated clients	Quantitative—surveys	7
	Qualitative—IDIs and FGDs	7
	Health care provider observation	2
	Facility audit	5
	Health records review	2

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
