# Peer review of "Exploring and Monitoring Privacy, Confidentiality, and Provider Bias in Sexual and Reproductive Health Service Provision to Young People: A Narrative Review"

_ijerph, 2022, doi:10.3390/ijerph19116576_

Round 1

Reviewer 1 Report

This is a wonderful manuscript. I really enjoyed reading the content.  This is an interesting and important study that highlighting poor privacy and confidentiality practices and provider bias serve as key barriers to care access for adolescent and young people.

I offer a few minor suggestions that may help your manuscript:

  • Why authors selected a narrative review rather than a systematic review?
  • I am not sure about the word “understanding” in the title, because I am not sure if this paper answer it
  • Is there some evidence in the COVID-19 context? 
  • It’s necessary doble check some size and font , for example the paragraph before 3.2 section
  • There is grey color in Appendix 1

Author Response

We would like to thank you for reviewing the manuscript, “Understanding and monitoring privacy, confidentiality, and provider bias in sexual and reproductive health service provision to young people: A narrative review.” Your comments have served to further improve the quality of this manuscript.

  1. Explain why the authors selected a narrative review rather than a systematic review.
    • An explanation has been included at the beginning of the methods section.
  2. Reconsider the use of the word “understanding” in the title.
    • We have changed the title to “Exploring and monitoring privacy, confidentiality, and provider bias in sexual and reproductive health service provision to young people: A narrative review.”
  3. Describe evidence of the effects of COVID-19 on sexual and reproductive health service access for to adolescent and young adult clients? 
    • While COVID-19 has likely served only to further restrict access to SRH services for young people, the literature reviewed for this manuscript mostly predates the COVID-19 pandemic.
  4. Doublecheck size and font, for example the paragraph before 3.2 section.
    • It appears that editors have fixed these issues.
  5. There is grey color in Appendix 1
    • This formatting change has been made.

Reviewer 2 Report

Congratulations to the authors for this complete and precise paper.

The paper is clearly presented and well organized. 

The Introduction extensively provide a theoretical and contextual framework of the issue.

Results and Discussion are coherent and clearly presented.

A possible suggestion in regard of the Method section:

The authors could specify more how they selected the papers to consider for review. Did only one author do this selection? Did two or more authors did separately the selection and then compared and discussed their selection in order to reach a final agreement?

Did the authors used the PRISMA method?

Finally, as regards Figure 2: what the different colors stand for? What the number within the country stand for? The authors could specify this aspect

Author Response

We would like to thank you for reviewing the manuscript, “Understanding and monitoring privacy, confidentiality, and provider bias in sexual and reproductive health service provision to young people: A narrative review.” Your comments have served to further improve the quality of this manuscript. 

  1. Specify how the authors selected the papers to consider for review. Did only one author do this selection? Did two or more authors separately the selection and then compare and discussed their selection in order to reach a final agreement?
    • Thank you for this suggestion. We have elaborated on the evidence selection and synthesis process at the end of the methods section.
  2. Did the authors used the PRISMA method?
    • As the review was narrative, rather than systematic in nature the authors did not strictly follow PRISMA guidelines. However, many of the reporting requirements of PRISMA were adopted for this review.
  3. In regard to Figure 2, explain the significance of the different colors and numbers signify.
    • The numbers and color shading correspond to the number of studies conducted within a country. Darker shades of green represent greater numbers of studies. This explanation has been included below the figure name.

Reviewer 3 Report

Here they are my comments:

Under the methods section, you need to elaborate on the narrative review performed and the reason for it in comparison with other types of review. Also, why do not you call it a systematic review? 

Information on the selction process of search results, evaluation process, research synthesis and data analysis etc are missing. Even if, this is not a systematic review, such details are needed to understand about the bases of your review, and the review results.

I suggest to include the table you assigned as appendix, in the text.

A figure or table would help with summarising your review results.

As with conclusion, what should be done for policy making and education to improve the current condition? 

Author Response

We would like to thank you for reviewing the manuscript, “Understanding and monitoring privacy, confidentiality, and provider bias in sexual and reproductive health service provision to young people: A narrative review.” Your comments have served to further improve the quality of this manuscript. 

  1. Under the methods section, elaborate on the narrative review performed and the reason for it in comparison with other types of reviews. Why is it not a systematic review? 
    • A statement regarding the merits of conducting a narrative review has been included in the beginning of the methods section.
  2. Information on the selection process of search results, evaluation process, research synthesis and data analysis etc. are missing.
    • Thank you for drawing our attention to this. We have elaborated on the selection and evidence synthesis process at the end of the methods section.
  3. Include appendices 1 and 2 as tables within the text.
    • These tables have been moved into the body of the article.
  4. Consider adding a figure or table that summarizes the review’s results.
    • The authors believe that the conclusion section offers a summary of the review’s key results.
  5. Include in the conclusion policy and practice considerations.
    • We have added reference to additional practice improvement resources in the discussion sections. The authors would also note that policy and practice considerations are considered in the results sub-sections titled, “strategies to improve privacy and confidentiality”, and “strategies to mitigate provider bias.”

Round 2

Reviewer 3 Report

Nothing more.